# Angiotensin I-Converting Enzyme (ACE) Inhibition and Molecular Docking Study of Meroterpenoids Isolated from Brown Alga, *Sargassum macrocarpum*

**DOI:** 10.3390/ijms241311065

**Published:** 2023-07-04

**Authors:** Seok-Chun Ko, Ji-Yul Kim, Jeong Min Lee, Mi-Jin Yim, Hyun-Soo Kim, Gun-Woo Oh, Chul Hwan Kim, Nalae Kang, Soo-Jin Heo, Kyunghwa Baek, Dae-Sung Lee

**Affiliations:** 1National Marine Biodiversity Institute of Korea, Seocheon 33662, Republic of Korea; seokchunk@mabik.re.kr (S.-C.K.); lshjm@mabik.re.kr (J.M.L.); mjyim@mabik.re.kr (M.-J.Y.); gustn783@mabik.re.kr (H.-S.K.); ogwchobo@mabik.re.kr (G.-W.O.); kchulh1204@mabik.re.kr (C.H.K.); kyunghwabaek@mabik.re.kr (K.B.); 2Jeju Marine Research Center, Korea Institute of Ocean Science and Technology (KIOST), Jeju 63349, Republic of Korea; nalae1207@kiost.ac.kr (N.K.); sjheo@kiost.ac.kr (S.-J.H.)

**Keywords:** *Sargassum macrocarpum*, meroterpenoids, angiotensin I-converting enzyme, molecular docking

## Abstract

Angiotensin I-converting enzyme (ACE) is an important blood pressure regulator. In this study, we aimed to investigate the ACE-inhibitory effects of meroterpenoids isolated from the brown alga, *Sargassum macrocarpum*, and the molecular mechanisms underlying ACE inhibition. Four fractions of *S. macrocarpum* were prepared using hexane, chloroform, ethyl acetate, and water as solvents and analyzed for their potential ACE-inhibitory effects. The chloroform fraction showed the strongest ACE-inhibitory effect, with an IC_50_ value of 0.18 mg/mL. Three meroterpenoids, sargachromenol, 7-methyl sargachromenol, and sargaquinoic acid, were isolated from the chloroform fraction. Meroterpenoids isolated from *S. macrocarpum* had IC_50_ values of 0.44, 0.37, and 0.14 mM. The molecular docking study revealed that the ACE-inhibitory effect of the isolated meroterpenoids was mainly attributed to Zn-ion, hydrogen bonds, pi-anion, and pi–alkyl interactions between the meroterpenoids and ACE. These results suggest that *S. macrocarpum* could be a potential raw material for manufacturing antihypertensive nutraceutical ingredients.

## 1. Introduction

Hypertension is a chronic health problem related to cardiovascular diseases, including myocardial infarction, arteriosclerosis, and stroke, and is affected by various factors, such as salt intake, smoking, stress, and obesity [1]. Among the processes associated with hypertension, angiotensin I-converting enzyme (ACE), which is involved in the kallikrein–kinin system (KKS) and renin–angiotensin system (RAS), plays a pivotal role in the development of hypertension [2,3,4]. ACE is a zinc metalloproteinase [5]. It is found in the vascular, heart, lung, and brain tissues [6]. In the KKS, ACE plays an important role in degrading the potent vasodilator bradykinin, while in the RAS, ACE catalyzes the conversion of the potent vasoconstrictor octapeptide angiotensin II (Asp-Arg-Val-Tyr-Ile-His-Pro-Phe) from the inactive decapeptide angiotensin I (Asp-Arg-Val-Tyr-Ile-His-Pro-Phe-His-Leu) [7]. Controlling blood pressure control by inhibiting ACE activity is regarded as a preventive and therapeutic strategy for hypertension. The synthetic ACE inhibitors, including captopril, enalapril, ramipril, and benazepril, are extensively used for the treatment of essential hypertension in humans [6,8,9]. However, these synthetic inhibitors have side effects, such as taste disturbances, skin rashes, and coughing [10,11]. Therefore, studies are focusing on using natural resources as potential ACE-inhibitory agents against hypertension.

Natural resources as alternative treatments for hypertension are gaining attention, and studies on herb-derived compounds are ongoing [12,13]. Marine algae possess various bioactive substances with potential pharmaceutical and nutraceutical properties [13]. Marine algae with bioactive substances differ from terrestrial plants [14]. Their antioxidative [15], antidiabetic [16], antiviral [17], anti-obesity [18], and antihypertensive [19] properties have been previously reported. Among brown algae, *Sargassum* species contain a variety of active compounds such as sterols, sulfated polysaccharides, polyphenols, meroterpenoids, and terpenoids [20], which possess anti-inflammatory [21], anticancer [22], antimicrobial [23], advanced glycation end product (AGE)-inhibitory [24], and neuroprotective [25] properties. Studies have shown that meroterpenoids contain sargachromenol, sargaquinoic acid, marcrocarquinoids A-C, and tuberatolide B, which possess biological activities [20,26]. Despite the potential use of the functional properties of meroterpenoids, to date, to the best of our knowledge, no study has reported their ACE-inhibitory effects and molecular mechanisms.

This study aimed to isolate meroterpenoids from the brown alga, *Sargassum macrocarpum* based on the analytical data results and evaluate their ACE-inhibitory effects and the molecular mechanism underlying ACE inhibition.

## 2. Results and Discussion

### 2.1. ACE Inhibitory Effect of S. macrocarpum Extracts

The inhibition of ACE activity can be a useful strategy for improving hypertension. Recently, various marine bioresources were studied for their ability to inhibit ACE activity to explore their applications in improving hypertension. In this study, extracts and five solvent fractions of *S. macrocarpum* were tested for their potential ACE-inhibitory effects. The 80% aqueous methanol extract of *S. macrocarpum* inhibited ACE activity in a dose-dependent manner by 6.1, 15.7, 40.7, 62.7, and 74.5% at concentrations of 0.0625, 0.125, 0.25, 0.5, and 1 mg/mL, respectively (Figure 1). The IC_50_ value of the extract against ACE was 0.38 mg/mL. Compared to the findings in a previous study, the ACE inhibitory effects of *S. macrocarpum* extract were more effective than those of the extracts from other Sargassum species [26]. This indicates that active compounds of different polarities, such as meroterpenoids, might be present in the *Sargassum* species, particularly in *S. macrocarpum*. Thus, *S. macrocarpum* was partitioned with hexane, chloroform, ethyl acetate, butanol, and water to detect the bioactive compounds, and their ACE-inhibitory effects were evaluated (Figure 2). The chloroform fraction showed the highest ACE-inhibitory effect with the lowest IC_50_ value 0.18 mg/mL compared to that of the other organic solvent fractions. Previous studies have shown that ACE-inhibitory compounds can be obtained from brown algae, such as *Ishige sinicola* [19], *Ecklonia cava* [27], and *Turbinaria ornata* [28]. Compared to the findings in our previous study, the ACE-inhibitory effect of the chloroform fraction of *S. macrocarpum* was more effective than that of the extract and chloroform fraction from *E. cava* [27].

### 2.2. Isolation and Identification of the Compounds

Next, the chloroform fraction was subjected to silica gel electrophoresis and MPLC. Finally, the compounds in this fraction were isolated using preparative HPLC and identified as meroterpenoids based on a comparison between the NMR spectroscopic data and previous literature [29,30]. Sargachromenol (1): brown oils; ^1^H NMR (500 MHz, CDCl_3_): δ_H_ 6.46 (1H, brs, H-7), 6.30 (1H, brs, H-5), 6.22 (1H, d, *J* = 9.8 Hz, H-4), 5.96 (1H, t, *J* = 7.0 Hz, H-7′), 5.55 (1H, d, *J* = 9.8 Hz, H-3), 5.12 (1H, t, *J* = 6.6 Hz, H-3′), 5.07 (1H, t, *J* = 6.4 Hz, H-11′), 2.58 (2H, m, H-6′), 2.24 (2H, m, H-9′), 2.11 (3H, s, H-18′), 2.11 (2H, m, H-2′ and H-10′), 2.05 (2H, m, H-5′), 1.65 (3H, s, H-13′), 1.64 (2H, m, H-1′), 1.56 (3H, m, H-14′), 1.55 (3H, m, H-16′), 1.33 (3H, m, H-17′) (Appendix A); ^13^C NMR (125 MHz, CDCl_3_): δ_C_ 173.4 (C-15′), 148.7 (C-6), 145.8 (C-7′), 144.9 (C-8a), 134.5 (C-4′), 132.4 (C-12′), 130.8 (C-3), 130.7 (C-8′), 126.5 (C-8), 125.0 (C-3′), 123.6 (C-11′), 123.1 (C-4), 121.5 (C-4a), 117.2 (C-7), 110.5 (C-5), 77.9 (C-2), 40.9 (C-1′), 39.2 (C-5′), 34.7 (C-9′), 28.3 (C-6′), 28.0 (C-10′), 26.0 (C-17′), 25.8 (C-13′), 22.8 (C-2′), 17.9 (C-14′), 15.9 (C-18′), 15.7 (C-16′) (Appendix A); HRESIMS: *m*/*z* 423.2539 C_27_H_35_O_4_ [M-H]^−^ (calcd for C_27_H_35_O_4_, 423.2535) (Appendix A).

7-methyl sargachromenol (2): brown oils; ^1^H NMR (500 MHz, CDCl_3_): δ_H_ 6.29 (1H, s, H-5), 6.21 (1H, d, *J* = 9.7 Hz, H-4), 5.97 (1H, s, H-7′), 5.52 (1H, d, *J* = 9.7 Hz, H-3), 5.11 (1H, t, *J* = 6.8 Hz, H-3′), 5.06 (1H, t, *J* = 6.7 Hz, H-11′), 2.58 (2H, m, H-6′), 2.23 (2H, m, H-9′), 2.12 (2H, m, H-2′ and H-10′), 2.11 (3H, s, H-19′), 2.10 (3H, s, H-18′), 2.05 (2H, m, H-5′), 1.65 (2H, m, H-1′), 1.65 (3H, s, H-13′), 1.56 (3H, s, H-14′), 1.55 (3H, s, H-16′), 1.33 (3H, s, H-17′) (Appendix A); ^13^C NMR (125 MHz, CDCl_3_): δ_C_ 172.6 (C-15′), 147.0 (C-6), 145.7 (C-7′), 144.9 (C-8a), 134.5 (C-4′), 132.4 (C-12′), 130.6 (C-8′), 130.0 (C-3), 128.3 (C-8), 125.2 (C-3′), 123.7 (C-11′), 123.6 (C-7), 123.0 (C-4), 118.9 (C-4a), 110.0 (C-5), 77.8 (C-2), 40.9 (C-1′), 39.2 (C-5′), 34.7 (C-9′), 28.3 (C-6′), 28.0 (C-10′), 26.0 (C-17′), 25.8 (C-13′), 22.8 (C-2′), 17.9 (C-14′), 15.9 (C-16′), 12.3 (C-19′), 11.8 (C-18′) (Appendix A); HRESIMS: *m*/*z* 437.2691 C_28_H_37_O_4_ [M-H]^−^ (calcd for C_28_H_37_O_4_, 437.2691) (Appendix A).

Sargaquinoic acid (3): brown oils; ^1^H NMR (500 MHz, _CDCl3_): δ_H_ 6.53 (1H, s, H-5), 6.44 (1H, s, H-1), 5.98 (1H, t, *J* = 7.2 Hz, H-10′), 5.13 (1H, m, H-2′), 5.11 (1H, m, H-6′), 5.07 (1H, m, H-14′), 3.11 (2H, m, *J* = 7.0 Hz, H-1′), 2.58 (2H, m, H-9′), 2.24 (2H, d, *J* = 7.4 Hz, H-12′), 2.10 (2H, m, H-13′), 2.07 (2H, m, H-5′), 2.05 (2H, m, H-4′ and H-8′), 2.04 (3H, s, H-21′), 1.65 (3H, s, H-17′), 1.60 (3H, s, H-20′), 1.58 (3H, s, H-19′), 1.56 (3H, s, H-16′) (Appendix A); ^13^C NMR (125 MHz, CDCl_3_): δ_C_ 188.3 (C-3), 188.2 (C-6), 172.7 (C-18′), 148.7 (C-2), 146.1 (C-4), 145.6 (C-10′), 140.0 (C-3′), 134.8 (C-7′), 133.3 (C-5), 132.49 (C-1), 132.44 (C-15′), 130.7 (C-11′), 124.6 (C-6′), 123.6 (C-14′), 118.1 (C-2′), 39.7 (C-4′), 39.2 (C-8′), 34.7 (C-12′), 28.4 (C-9′), 28.0 (C-13′), 27.7 (C-1′), 26.5 (C-5′), 25.9 (C-17′), 17.9 (C-16′), 16.3 (C-21′), 16.2 (C-19′), 16.1 (C-20′) (Appendix A); HRESIMS: *m*/*z* 423.2534 C_27_H_35_O_4_ [M-H]^−^ (calcd for C_27_H_35_O_4_, 423.2535) (Appendix A).

### 2.3. Inhibitory Effects and Molecular Docking of Compounds Isolated from S. macrocarpum to ACE

Three meroterpenoids, sargachromenol, 7-methyl sargachromenol, and sargaquinoic acid, with different chemical structures and molecular weights were isolated. The ACE-inhibitory effects of the three meroterpenoids are shown in Figure 3. Among the meroterpenoids isolated, sargaquinoic acid showed the highest effect compared to that of the other meroterpenoids. In terms of the activation of the ACE-inhibitory effect, the lowest IC_50_ value was observed for sargaquinoic acid at a concentration of 0.14 mM, and IC_50_ value of sargaquinoic acid was 3-fold lower than that of sargachromenol, which was the weakest inhibitor among the three meroterpenoids. It has previously been found that plant phenolic compounds such as quercetin, kaempferol, rutin, apigenin, epicatechin, phloretin, and resveratrol exhibit ACE-inhibitory effects and have IC_50_ values of 0.415, 0.512, 0.472, 0.667, 1.381, 1.110, and 0.970 mM, respectively [31]. Additionally, they are ACE-inhibitory compounds derived from marine plants. The isolated ACE-inhibitory compounds are phloroglucinol (IC_50_ value = 2.57 mM), triphlorethol-A (IC_50_ value = 2.01 mM), eckol (IC_50_ value = 2.27 mM), dieckol (IC_50_ value = 1.47 mM), and eckstolonol (IC_50_ value = 2.95 mM) [27]. In this study, the isolated meroterpenoids exhibited stronger or similar effects, but demonstrated weaker effects than that of several ACE-inhibitory peptides derived from various natural plants. The identified ACE-inhibitory peptides are Asp-Glu-Asn-Ser-Lys-Phe (IC_50_ value = 0.1 mM) from Terminalla chebula tree [32], Tyr-Ser-Lys (IC_50_ value = 0.076 mM) from rice bran [33], Val-Ser-Gly-Ala-Gly-Arg-Tyr (IC_50_ value = 0.0086 mM) from bitter melon seed [34], and Lys-Glu-Asp-Asp-Glu-Glu-Glu-Glu-Gln-Glu-Glu-Glu (IC_50_ value = 0.064 mM) from Pea [35]. A previous study demonstrated that meroterpenoids isolated from *S. macrocarpum* exhibited a significantly weaker effect than that of lisinopril (IC_50_ value = 2.2 nM), enalapril (IC_50_ value = 6.3 nM), and captopril (IC_50_ value = 6.3 nM), currently the most widely used synthetic ACE-inhibitor [36]. Diet therapy through natural products with fewer undesirable side-effects and lower cost is safer than medicines. Therefore, the natural products may become an important component in the initial treatment of patients with mild hypertension. However, the stable forms of bioactive molecules and their specific doses in humans are difficult to determine and commercialize. It is thought that additional studies on the bioavailability of natural products and studies on the effects that may occur when taken together with commercial ACE inhibitor are needed. To explore the structural–functional interactions between the meroterpenoids (sargachromenol, 7-methyl sargachromenol, and sargaquinoic acid) and ACE, computational molecular docking was performed using Discovery Studio 2022. The C-terminal tripeptide of substrate has been reported to affect ACE binding [37,38]. Additionally, the three marine active site pockets (S1, S’1, and S’2) of ACE play a crucial role in the interaction between the inhibitor and ACE [37]. S1 (antepenultimate) contains ALA354, GLU384, and TYR523 residues; S’1 contains GLU162; and S2 contains GLN281, HIS353, HIS513, and LYS511 [39]. Lisinopril, an ACE inhibitor used for the clinical treatment of hypertension, interacts with the S1 pocket (ALA354 and GLU384) and S2 pocket (HIS353) [37]. The most stable meroterpenoid bond with ACE was obtained, and the 2D and 3D structures are shown in Figure 4. Among the meroterpenoids, sargachromenol and 7-methyl-sargachromenol do exist as *R*- and *S*-isomers, and structures of meroterpenoids are described in Table 1. Sargachromenol *R*-isomer interacted with GLU376 (conventional hydrogen bond interaction), ALA354 (alkyl interaction), VAL379 (alkyl interaction), VAL380 (alkyl interaction), VAL518 (alkyl interaction), PHE527 (Pi-alkyl interaction), HIS353 (Pi-alkyl interaction), HIS383 (Pi-alkyl and carbon hydrogen bond interaction), HIS513 (Pi-alkyl interaction), PHE457 (Pi-alkyl interaction), and TYR523 (Pi-alkyl interaction) (Figure 4A). Sargachromenol *S*-isomer interacted with ALA354 (alkyl interaction), VAL380 (alkyl interaction), VAL518 (alkyl interaction), HIS353 (Pi-alkyl interaction), HIS383 (carbon hydrogen bond interaction), HIS513 (Pi-alkyl interaction), HIS387 (Pi-alkyl and Pi-anion interaction), and TRP279 (Pi-alkyl interaction) (Figure 4B). 7-methyl-sargachromenol *R*-isomer interacted with TYR523 (conventional hydrogen bond interaction), GLU376 (conventional hydrogen bond interaction), VAL380 (alkyl interaction), LEU161 (alkyl interaction), ALA354 (alkyl interaction), HIS353 (Pi-alkyl interaction), PHE512 (Pi-alkyl interaction), and TRP279 (Pi-alkyl interaction) (Figure 4C). 7-methyl-sargachromenol *S*-isomer interacted with TYR523 (conventional hydrogen bond interaction), LEU161 (alkyl interaction), HIS353 (Pi-alkyl interaction), HIS387 (Pi-alkyl interaction), HIS410 (Pi-alkyl interaction), HIS513 (Pi-alkyl interaction), PHE512 (Pi-alkyl interaction), TRP279 (Pi-alkyl interaction), ALA354 (alkyl interaction), GLU411 (carbon hydrogen bond interaction), CYS352 (alkyl interaction), and CYS370 (alkyl interaction) (Figure 4D). In the case of sargaquinoic acid, the binding site was formed by the following: CYS370 (conventional hydrogen bond interaction), ALA354 (alkyl interaction), VAL380 (alkyl interaction), VAL518 (alkyl interaction), PRO163 (alkyl interaction), HIS353 (Pi-alkyl interaction), HIS383 (Pi-alkyl interaction and carbon hydrogen bond), HIS387 (Pi-alkyl interaction), PHE512 (Pi-alkyl interaction), TYR523 (Pi-alkyl interaction), GLU162 (Pi-anion), ASP377 (Pi-anion), and SER355 (carbon hydrogen bond interaction) (Figure 4E). The molecular docking analysis indicated that the computed binding energy (kcal/mol) values of sargachromenol *R*-isomer, sargachromenol *S*-isomer, 7-methyl sargachromenol *R*-isomer, 7-methyl sargachromenol *S*-isomer, and sargaquinoic acid with ACE were −198.931, −190.337, −218.374, −227.35 and −248.169 kcal/mol, respectively (Table 2). Among the ACE active sites, Zn^2+^ is involved in ACE activation and binds to the ACE residues GLU411, HIS383, and HIS387 [40]. The molecular docking study revealed that meroterpenoids did interact directly with Zn^2+^ at the active site that sargachromenol *R*-isomer forms with S1 (ALA354 and TYR523) and S2 (HIS353 and HIS513) active site pockets and that sargachromenol *S*-isomer forms with S1 (ALA354) and S2 (HIS353, and HIS513) active site pockets. In the case of 7-methyl sargachromenol, *R*-isomer forms with S1 (TYR523) and S2 (HIS353) active site pockets of ACE, and *S*-isomer forms with S1 (ALA 354, and TYR523) and S2 (HIS353, and HIS513) active site pockets of ACE. In addition, sargaquinoic acid forms an active site with the S1 (ALA354, TYR523), S’1 (GLU162), and S2 (HIS353) active site pockets of ACE.

In conclusion, among the various organic solvent fractions, the chloroform fraction exhibited the highest ACE inhibitory effect. Three meroterpenoids, sargachromenol, 7-methyl sargachromenol, and sargaquinoic acid, having ACE-inhibitory effects were isolated, with IC_50_ values 0.44, 0.37, and 0.14 mM, respectively. Among the three meropterpenoids, sargachromenol and 7-methyl-sargachromenol do exist as *R*- and *S*-isomers, and molecular docking studies of the respective other isomers were performed. These other isomers showed the similar docking poses and binding energies with those of isomers. Molecular docking studies revealed that these meroterpenoids in the ACE active site contributed to the stabilization of the docking complex, and it was confirmed that sargaquioic acid contributed the most to stabilization. These findings provide a partial molecular explanation for the ACE-inhibitory properties of meroterpenoids isolated from *S. macrocarpum*. Moreover, meroterpenoids are promising for the discovery of potential candidates for the future industrial production of functional foods.

## 3. Materials and Methods

### 3.1. Materials

*S. macrocarpum* was collected from Jeju Island, South Korea. The alga was washed three times with tap water to remove salt and epiphytes attached to the surface, followed by careful rinsing with fresh water. The algae were stored in a medical refrigerator at −80 °C. Frozen algae were lyophilized and homogenized using a grinder prior to extraction. The ACE assay kit (ACE kit-WST) was obtained from Dojindo Laboratories (Kumamoto, Japan).

### 3.2. Extraction and Isolation of Compounds

The dried *S. macrocarpum* powder was extracted with 80% aqueous methanol at room temperature, and the liquid layer was obtained via filtration. The filtrate was evaporated to obtain the methanol extract, which was suspended in distilled water and partitioned with hexane, chloroform, ethyl acetate, and butanol. The chloroform fraction exhibited a stronger ACE-inhibitory effect than the other fractions. Thus, the compounds were extracted from the chloroform fraction using a silica gel column and medium-pressure liquid chromatography (Selekt MPLC system, biotase, Uppsala, Sweden). Finally, the compounds were purified using preparative high-performance liquid chromatography (Nextra prep-HPLC, Shimadzu, Kyoto, Japan), and the structures of the compounds were identified by comparing the nuclear magnetic resonance (Varian VNMRS 500 MHz FT-NMR spectrometer, Varian, Pala Alto, CA, USA) spectral data and high-resolution electrospray ionization mass spectrometry (SCIEX X500R Q-TOF LC-MS/MS spectrometer, SCIEX, Framingham, MA, USA) analysis results with those in the existing literature.

### 3.3. Determining the ACE Inhibitory Activity

The ACE-inhibitory activities of the extract, fractions, and compounds were determined using an ACE kit-WST (Dojindo Laboratories). The assay was performed according to the manufacturer’s instructions. The absorbance was measured at 450 nm using a microplate reader (Model 550, Bio-Rad, Hercules, CA, USA). IC_50_ value was defined as the concentration of extract and fractions (mg/mL), and compounds (µM) required for 50% reduction of the ACE activity.

### 3.4. Three-Dimensional (3D) Structure of the Proteins and Compounds

Molecular docking was performed using CDOCKER and Calculate Binding Energies tools in Discovery Studio 2022 (Biovia, San Diego, CA, USA) to assess the binding positions of meroterpenoids within the active site of ACE. A docking mechanism based on CHARMm [41] was used to execute the CDOCKER docking protocol. The docking of marine aldehyde derivatives to the ACE was performed as follows: (1) a two-dimensional (2D) structure was converted to a 3D structure, (2) proteins were prepared, and the binding sites were defined, and (3) the compounds were docked [42]. The ACE-binding pocket was defined as the area from the center of the active site up to a radius of 11.2 Å. The binding site and ligand were allowed to move freely during the docking. Water molecules were removed from the protein in the flexible docking process because the fixed water molecules might alter the generation of the ligand–receptor complex. After the removal of water molecules, hydrogen atoms were attached to the protein. Ligand binding affinity was assessed for all complexes by applying the CHARMm force field to the interaction energy. Based on CDOCKER’s interaction energy, the distinct conformational positions for each molecule were identified and examined. The binding energies of the small-molecule protein complexes were calculated using the Calculated Binding Energies tool and used to select candidate compounds. The docking positions of the selected compounds to ACE are expressed as 2D and 3D crystal structures.

### 3.5. Statistical Analyses

Data are presented as mean ± standard error. Statistical comparisons of the mean values were performed using analysis of variance, followed by Duncan’s multiple-range test using the SPSS software (Version 21). A *p* < 0.05 was considered statistically significant.

## Figures and Tables

**Figure 1 ijms-24-11065-f001:**
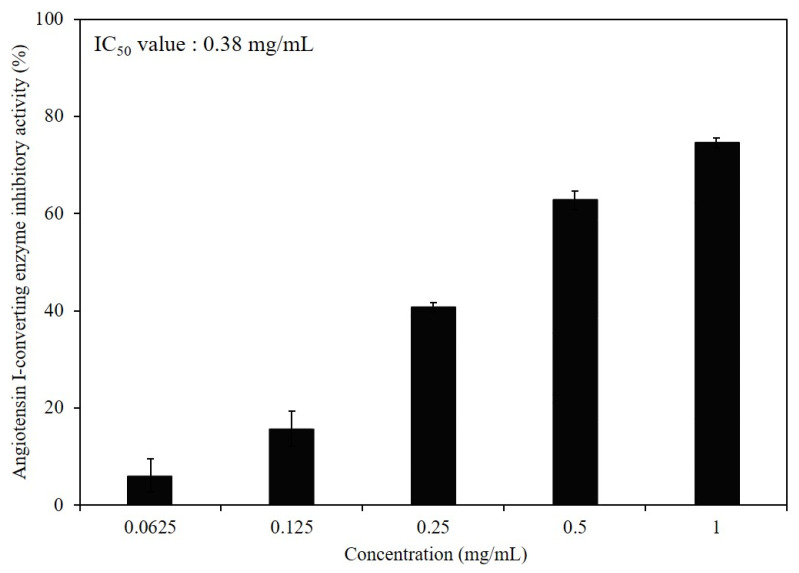
Angiotensin I-converting enzyme (ACE) inhibitory effect of extract from *Sargassum macrocarpum*. Values are expressed as mean ± S.D. of triplicate experiments. IC_50_ value is the concentration of extract required for 50% inhibition.

**Figure 2 ijms-24-11065-f002:**
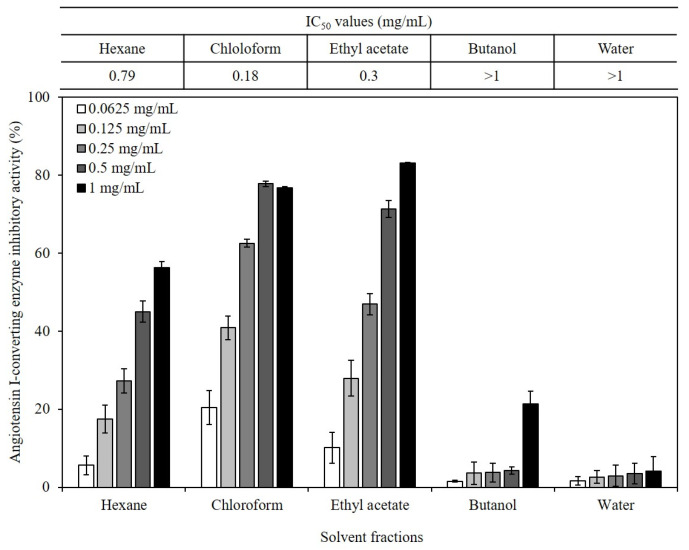
Angiotensin I-converting enzyme (ACE) inhibitory effect of fractions from *Sargassum macrocarpum* extract. Values are expressed as mean ± S.D. of triplicate experiments. IC_50_ value is the concentration of fractions required for 50% inhibition.

**Figure 3 ijms-24-11065-f003:**
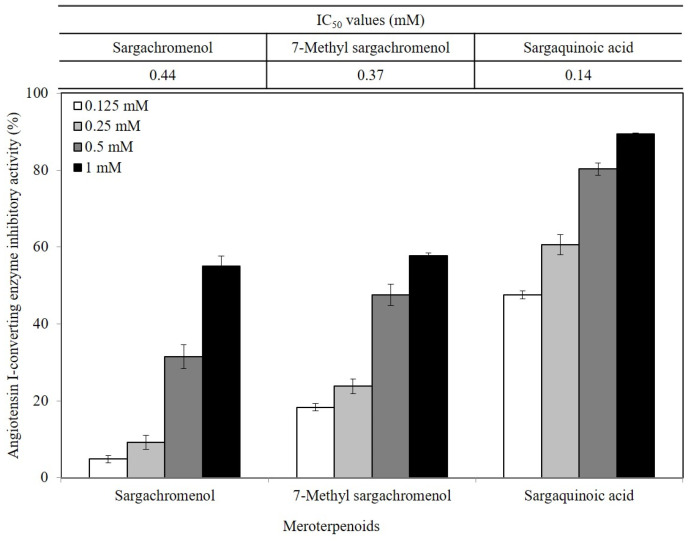
Angiotensin I-converting enzyme (ACE) inhibitory effect of meroterpenoids from *Sargassum macrocarpum*. Values are expressed as mean ± S.D. of triplicate experiments. IC_50_ value is the concentration of meroterpenoids required for 50% inhibition.

**Figure 4 ijms-24-11065-f004:**
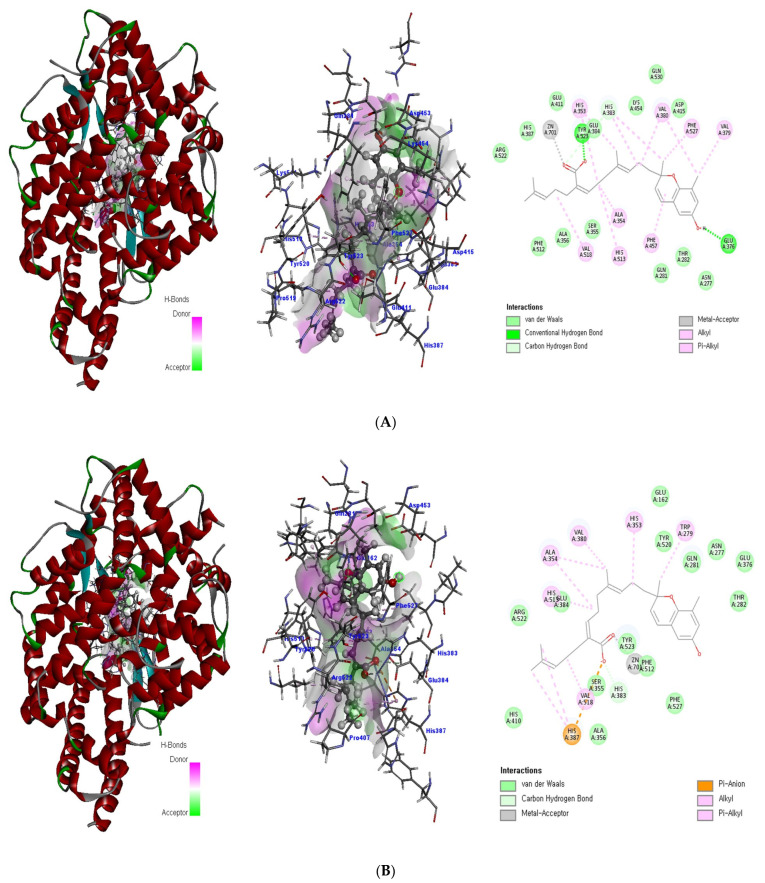
The docking poses of the meroterpenoid-active site of ACE. The 3D and 2D diagram of the complexes to active site with sargachromenol *R*-isomer (**A**), sargachromenol *S*-isomer (**B**), 7-methyl sargachromenol *R*-isomer (**C**), 7-methyl sargachromenol *S*-isomer (**D**), and sargaquinoic acid (**E**). The active site was expressed as a ribbon model tagging the amino acid. The meroterpenoids are shown as a gray and red stick model, and the binding surface is expressed in terms of hydrogen bonds. The 2D diagram of the meroterpenoids-active site complexes were combined as hydrogen bond or pi bond.

**Table 1 ijms-24-11065-t001:** 2D and 3D structure of the meroterpenoids.

Compounds	Structures
*R*-sargachromenol	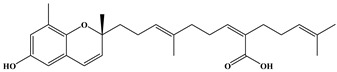
*S*-sargachromenol	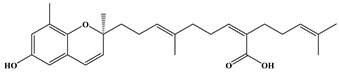
*R*-7-methyl sargachromenol	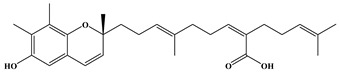
*S*-7-methyl sargachromenol	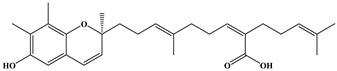
sargaquinoic acid	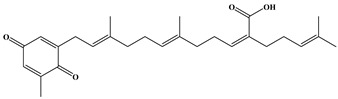

**Table 2 ijms-24-11065-t002:** The computed binding energy values of the meroterpenoids.

Meroterpenoids	Computed Binding Energy (kacal/mol)
*R*-Sargachromenol	−198.931
*S*-Sargachromenol	−190.337
*R*-7-methyl sargachromenol	−218.374
*S*-7-methyl sargachromenol	−227.35
Sargaquinoic acid	−248.169

## Data Availability

Not applicable.

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
