# Peer review of "Angiotensin I-Converting Enzyme (ACE) Inhibition and Molecular Docking Study of Meroterpenoids Isolated from Brown Alga, Sargassum macrocarpum"

_ijms, 2023, doi:10.3390/ijms241311065_

Round 1
Reviewer 1 Report
There are only some minor issues in this otherwise well written manuscript.
Minor issues:
(1) Introduction line 39: Currently, also ramipril is widely used.
(2) Sargachromenol and 7-methyl-Sargachromenol do exist as R- and S-isomers.
From the depiction in Table 1 is not obvious to tell which isomer
was used for docking, which can of course make a huge difference.
Please clarify which isomer was used.
Probably you have to perform the docking for the respective other isomer
as well.
(3) The carboxylic acid groups of all compounds will be in their deprotonated form and thus directly interact with the Zn-ion.
Please check if this is the case. Otherwise the obtained binding poses are simply wrong.
(4) The magnitude of the reported "binding energies" in Table 2 is way too
high if compared with values computed from DeltaG = -RT ln K_i.
For example, enalapril would give rise to values around -10 kcal/mol.
Therefore I suggest to change the heading "binding energy" in Table 2
to "computed binding energy (kcal/mol)". Same for the text around line 138.
(4) I would recommend to drop the 3D-depiction from Table 1 and
instead indicate the stereochemistry in the 2D-depiction.
Somehow I don't get what the numbers next to the atoms mean.
1H-NMR and 13C-NMR chemical shifts? Please indicate in the header of
Table 1.
Author Response
Ijms-2436836
Angiotensin I-converting enzyme (ACE) inhibition and molecular docking study of meroterpenoids isolated from brown alga, Sargassum macrocarpum
International Journal of Molecular Sciences
Dear Sir.
Thank you for your kind comments and guidelines on the manuscript. Further, our corrections for the reviewer comments are in red color in the revised manuscript. Herewith I send you the revised manuscript with correction sheet.
Comment 1
Introduction line 39: Currently, also ramipril is widely used.
Response: According to your comment, we have mentioned the ramipril in Introduction section (Line 58).
Comment 2
Sargachromenol and 7-methyl-Sargachromenol do exist as R- and S-isomers. From the depiction in Table 1 is not obvious to tell which isomer was used for docking, which can of course make a huge difference. Please clarify which isomer was used. Probably you have to perform the docking for the respective other isomer as well.
Response: According to your comment, we have performed the docking for the R- and S-isomers of Sargachromenol and 7-methyl-Sargachromenol. And, we have added the docking data in Fig 4.
Comment 3
The carboxylic acid groups of all compounds will be in their deprotonated form and thus directly interact with the Zn-ion. Please check if this is the case. Otherwise the obtained binding poses are simply wrong.
Response: Your comments are greatly appreciated. We checked the structure of all compounds, ACE structure (1O86) as well as each complex, and we confirmed that the carboxylic acid group can directly bind with Zn-ion covalently bound to GLU411 of protein. Please consider that we used CDOCKER program in this report to refine and optimize docking poses and prioritize compounds although Genetic Optimization for Ligand Docking (GOLD) program is enable docking of covalently bound ligands to the receptor.
According to your comment, we tried to adjust the torsion angle of only carboxylic acid group, and present the simply revised biding pose of each complex to improve the binding poses.
Comment 4
The magnitude of the reported "binding energies" in Table 2 is way to high if compared with values computed from DeltaG = -RT ln K_i. For example, enalapril would give rise to values around -10 kcal/mol. Therefore I suggest to change the heading "binding energy" in Table 2 to "computed binding energy (kcal/mol)". Same for the text around line 138.
Response: Your comments are greatly appreciated. Based on your comment, we revised the “binding energy” to “computed binding energy (kcal/mol)” throughout the manuscript (Line 181).
Comment 5
I would recommend to drop the 3D-depiction from Table 1 and instead indicate the stereochemistry in the 2D-depiction. Somehow I don't get what the numbers next to the atoms mean. 1H-NMR and 13C-NMR chemical shifts? Please indicate in the header of Table 1.
Response: According to your comments, we have deleted the 3D structures of sargachromenol, 7-methyl sargachromenol, sargaquinoic acid. Also, We have added the each 2D structures of compounds in Table 1 and added the assigned the 1H and 13C NMR spectroscopic data of compounds 1-3 in manuscript (Line 100 to 124).

Reviewer 2 Report
First, there are two points to add in this paper:
- The authors claim they have isolated 3 natural products but they don't provide any evidence for that: they must give in SI the copies of the 1H and 13C spectra of their isolated fractions so the reviewer/reader can have a proof.
- Now concerning the biological data: it is always required to perform in parallel biological studies with a known, reference compound. In that case it's easy since there are a large number of known and potent ACE inhibitors. Such data will help to guarantee the quality of the biological tests performed and also to "calibrate" the activity of the novel isolated products with known compounds.
In a more minor mode, there is a comprehensive review by Rushi, M., RSC Advances, 2020 10 (42) 24951-72 which should be mentionned.
For this paper, the major key isssue is in fact the significance of this work: since 40 years we have very potent ACE inhibitors which are quiet safe and widely used as drugs worldwide......The authors present here compounds with IC50 around 100 micromolar while the commercial drugs have IC50s in the low nanomolar range, that means with an affinity at least 100000 times higher . Therefore the ACE guideline (and the interest!!) of this research appears very low and at least from this paper, the reviewer don't see (at this level) a significant "added value" to these molecules regarding the therapeutics linked to ACE inhibitors.
However, since these are novel properties for these natural products this paper could be publisehed after addition of the two key aspects mentioned above and being a little more critical regarding the real interest of the ACE inhibition data for these compounds.
minor adjustments
Author Response
Ijms-2436836
Angiotensin I-converting enzyme (ACE) inhibition and molecular docking study of meroterpenoids isolated from brown alga, Sargassum macrocarpum
International Journal of Molecular Sciences
Dear Sir.
Thank you for your kind comments and guidelines on the manuscript. Further, our corrections for the reviewer comments are in red color in the revised manuscript. Herewith I send you the revised manuscript with correction sheet.
Comment 1
The authors claim they have isolated 3 natural products but they don't provide any evidence for that: they must give in SI the copies of the 1H and 13C spectra of their isolated fractions so the reviewer/reader can have a proof.
Response: According to your comments, we have added the 1D (1H-and 13C-) NMR spectra and HRESI-MS data of compounds 1-3 in supporting information.
Comment 2
Now concerning the biological data: it is always required to perform in parallel biological studies with a known, reference compound. In that case it's easy since there are a large number of known and potent ACE inhibitors. Such data will help to guarantee the quality of the biological tests performed and also to "calibrate" the activity of the novel isolated products with known compounds.
Response : Your comments are greatly appreciated. We have revised the manuscript according to your recommendation (Line 133 to 143).
Comment 3
In a more minor mode, there is a comprehensive review by Rushi, M., RSC Advances, 2020 10 (42) 24951-72 which should be mentionned.
Response : According to your comments, we have mentioned the review (Line 67 to 72, Reference NO. 20).
Comment 4
For this paper, the major key isssue is in fact the significance of this work: since 40 years we have very potent ACE inhibitors which are quiet safe and widely used as drugs worldwide......The authors present here compounds with IC50 around 100 micromolar while the commercial drugs have IC50s in the low nanomolar range, that means with an affinity at least 100000 times higher . Therefore the ACE guideline (and the interest!!) of this research appears very low and at least from this paper, the reviewer don't see (at this level) a significant "added value" to these molecules regarding the therapeutics linked to ACE inhibitors.
However, since these are novel properties for these natural products this paper could be publisehed after addition of the two key aspects mentioned above and being a little more critical regarding the real interest of the ACE inhibition data for these compounds.
Response: Your comments are greatly appreciated. We have revised the manuscript according to your recommendation (Line 143 to 151).

Round 2
Reviewer 2 Report
This paper has been improved but there is still a key issue: I asked the authors to provide in Supplementary Material copies of their 1H and 13C for all isolated compounds and they did not do it, even if it is just a starightforward process in scanning and copying their spectra. They must do it since it is a proof for the structure and putity of their isolated compounds!!! Instead of that, they gave the NMR data of these molecules but this cannot be considered as a proof since such data should be already available somewhere in the literature. The SI does not seem linked to the text and therefore it will not be available to the potential readers.
Nothing to add
Author Response
Comment 1
This paper has been improved but there is still a key issue: I asked the authors to provide in Supplementary Material copies of their 1H and 13C for all isolated compounds and they did not do it, even if it is just a starightforward process in scanning and copying their spectra. They must do it since it is a proof for the structure and putity of their isolated compounds!!! Instead of that, they gave the NMR data of these molecules but this cannot be considered as a proof since such data should be already available somewhere in the literature. The SI does not seem linked to the text and therefore it will not be available to the potential readers.
Response : When we first replied to the revision, we have sent the NMR data (SI) as an attachment, and attaching it again because there is a possibility that it may be missing.

Round 3
Reviewer 2 Report
With the modifications and the spectra in SI, it is suitable for publication now.
No special comment